# Diagnostic and Prognostic Potential of MiR-379/656 MicroRNA Cluster in Molecular Subtypes of Breast Cancer

**DOI:** 10.3390/jcm10184071

**Published:** 2021-09-09

**Authors:** Megha Lal, Asgar Hussain Ansari, Anurag Agrawal, Arijit Mukhopadhyay

**Affiliations:** 1Genomics & Molecular Medicine Unit, CSIR-Institute of Genomics & Integrative Biology, Delhi 110025, India; meghalal110@gmail.com (M.L.); asgar.hussain@igib.in (A.H.A.); a.agrawal@igib.in (A.A.); 2Academy of Scientific and Innovative Research, Ghaziabad, Uttar Pradesh 201002, India; 3Biomedical Research Centre, Translational Medicine Unit, University of Salford, Manchester M5 4WT, UK

**Keywords:** MiR-379/656, 14q32, breast cancer, prognosis, diagnosis, miRNA, biomarker

## Abstract

Introduction: Breast cancer is the most frequently diagnosed cancer globally and is one of the most important contributors to cancer-related deaths. Earlier diagnosis is known to reduce mortality, and better biomarkers are needed. MiRNA clusters often co-express and target mRNAs in a coordinated fashion, perturbing entire pathways; they thus merit further exploration for diagnostic or prognostic use. MiR-379/656, at chromosome 14q32, is the second largest miRNA cluster in the human genome and implicated in various malignancies including glioblastoma, melanoma, gastrointestinal tumors and ovarian cancer highlighting its potential importance. In this study, we focus on the diagnostic and prognostic potentials of MiR-379/656 in breast cancer and its molecular subtypes. Materials and Methods: We analyzed miRNA and mRNA next generation sequencing data from 903 primary tumors and 90 normal controls (source: The Cancer Genome Atlas). The differential expression profile between tumor and normal was analyzed using DeSEQ2. Penalized logistic regression modelling (lasso regression) was used to assess the predictive potential of MiR-379/656 expression for tumor and normal samples. The association between MiR-379/656 expression and overall patient survival was studied using Cox Proportional-Hazard Model. The target mRNAs (validated) of MiR-379/656 were annotated via pathway enrichment analysis to understand the biological significance of the cluster in breast cancer. Results: The differential expression analysis for 1390 miRNAs (miRnome) revealed 310 upregulated (22.3%) and 176 downregulated (12.66%) miRNAs in breast cancer patients compared with controls. For MiR-379/656, 32 miRNAs (32/42; 76%) were downregulated. The MiR-379/656 cluster was found to be the most differentially expressed cluster in the human genome (*p* < 10^−30^). The Basal and Luminal B subtypes showed at least 83% (35/42) of the miRNAs to be downregulated. The binomial model prioritized 15 miRNAs, which distinguished breast cancer patients from controls with 99.15 ± 0.58% sensitivity and 77.78 ± 5.24% specificity. Overall, the Basal and Luminal B showed the most effective predictive power with respect to the 15 prioritized miRNAs at MiR-379/656 cluster. The decreased expression of MiR-379/656 was found to be associated with poorer clinical outcome in Basal and Luminal B subtypes, increasing tumor stage and tumor size/extent, and overall patient survival. Pathway enrichment for the validated targets of MiR-379/656 was significant for cancer-related pathways, especially DNA repair, transcriptional regulation by p53 and cell cycle checkpoints (adjusted *p*-value < 0.05). Conclusions: Genome informatics analysis of high throughput data for MiR-379/656 cluster has shown that a subset of 15 miRNAs from MiR-379/656 cluster can be used for the diagnostic and prognostic purpose of breast cancer and its subtypes—especially in Basal and Luminal B.

## 1. Introduction

Breast cancer is characterized by uncontrolled proliferation in the lobules, ducts or in the fatty and the fibrous connective tissue within the breast. WHO administered International Agency for Research on Cancer (IARC) data show that breast cancer leads to the highest number of cancer cases and deaths in women [1]. This emphasizes the constant need of development of treatment and surveillance strategies for better disease management.

MiRNAs, 20–22 nucleotides in length, are epigenetic regulators that usually function as repressors of gene expression [2]. MiRNAs have been known to play important regulatory roles in disease pathogenesis, particularly in cancer [3,4,5]. Moreover, miRNAs affect many cancer-related processes including proliferation, cell cycle control, apoptosis, differentiation, migration, metabolism and stress response [6,7,8]. One of the first reports implicating the involvement of miRNAs in cancer came in early 2000 where miR-15a and miR-16-1 at 13q14 were found to be downregulated or deleted in approximately 68% of the patients suffering from B-cell chronic lymphocytic leukemia [9]. Since then, a number of miRNAs have been discovered to play important regulatory roles in different types of cancer including, breast, colon, gastric, lung, prostate and thyroid [10,11]. Besides, tissue specificity, stability and easy detection in bodily fluids such as blood, serum, and urine provide powerful opportunities for development of miRNA-based biomarkers in cancer diagnostics and therapeutics [12]. For example, a recent report showed that the expression of miR-23a-3p, miR-130a-5p, miR-144-3p, miR-148a-3p and miR-152-3p, in plasma, can be used as biomarkers for early diagnosis of breast cancer [13].

Distribution of miRNAs in the genome is not random, and genomic clusters are frequently seen [14,15]. There are currently 132 such genomic clusters that contain over 20% of known miRNAs [16]. MiRNA clusters are defined as miRNA genes located within 10 Kb of distance on the same chromosome and in the same strand of DNA [17]. MiRNA clusters are frequently co-regulated and co-expressed, targeting multiple mRNA/Protein within the same or similar pathways [18,19,20,21]. MiR-17~92 at chromosome 13q31 shows over-expression in different malignancies including breast cancer, lung cancer, B-cell lymphomas and acute lymphoblastic leukemia [22,23,24,25]. MiRNA members of MiR-17~92 target mRNAs in a coordinated manner to promote proliferation, increase angiogenesis and sustain cell survival [26,27,28]. Similarly, MiR-221/222 targets TRPS1, a member of the GATA family of transcriptional repressors in breast cancer. TRPS1 inhibits epithelial to mesenchymal transition (EMT) by directly inhibiting expression of ZEB2 [29]. MiR-99a~125b cluster represses many mRNAs of TP53, Erb and MAPK signaling pathways in multiple myeloma cells [30]. There has been extensive research on miRNAs implicated in cancers; however, studies on large miRNA clusters in cancer are rare. MiR-379/656 on chromosome 14q32 is the second largest miRNA cluster in humans (~45 Kb) and harbors 42 miRNA encoding genes. MiR-379/656 is situated within the imprinted domain, DLK1-DIO3, unique to placental mammal lineage, and shows enriched expression in brain and placenta [31]. The cluster plays an important role in growth and development [32] and is implicated in various malignancies [33,34,35,36,37] including breast cancer [34,38,39]. These reports have been based on a limited number of samples or a limited number of candidate miRNAs. To date, the diagnostic and prognostic potentials of the entire MiR-379/656 cluster as a biomarker in breast cancer and its subtypes remain unknown. We present a comprehensive analysis of miRNA and mRNA next generation sequencing data that addresses the clinical and biological relevance of MiR-379/656 in breast cancer and its molecular subtypes.

## 2. Materials and Methods

### 2.1. Genomic Annotation of Clustered miRNAs

The genomic locations of the miRNAs were downloaded from miRBase v18 [40]. MiRNAs within 10 kb of distance, on the same chromosome and on the same strand, were clustered together. We identified 431 miRNAs to be nested within 132 miRNA clusters (Appendix A).

### 2.2. Downloading and Preprocessing TCGA Data

The miRNA and mRNA expression data of TCGA BRCA and the associated clinical information were downloaded using the R package, TCGAbiolinks [41]. Samples with median tumor purity < 0.6 were filtered out, as per the recommended threshold [42]. There were 1870 miRNAs in total, which were measured across 993 samples. Of which, 480 miRNAs with total read count <10 were removed to improve the accuracy of the analysis (Appendix A). The PVCA module of R package, ExpressionNormalizationWorkflow [43] was used to determine the proportion of variance contributed by different sources of plausible biological (Appendix A) and technical variations (Appendix A). For each biological feature, the contributed variation was <10% (Appendix A). The residual variation was 86.5%, which might be due to inter-individual differences. In case of technical features, each confounding variables contributed to <10% variation in miRNA expression (Appendix A). The residual variation was 81.3%, which might be due to inter-individual differences. Survival data and molecular subtype information of TCGA BRCA patient samples were obtained from UCSC Xena browser [44].

### 2.3. Differential miRNA Expression Analysis in Breast Cancer

The curated miRNA expression matrix consisted of 1390 miRNAs and 993 samples (903 primary tumors and 90 normal controls). The primary tumors were further classified on the basis of their molecular subtypes (PAM50)—Basal (136), Her-2 (59), Luminal A (491) and Luminal B (179). Prior to differential expression analysis, the relationship between sample condition (tumor and normal) and different environmental variables was determined using Fisher’s exact test and T-test (Appendix A). It was observed that there was a significant difference between the number of tumor and normal samples sequenced on different Illumina platforms (*p*-value < 0.01). To determine if this association had an impact on miRNA expression, the data quality was assessed by sample clustering (Appendix A). The samples formed two distinct clusters with sample condition (tumor, normal). However, with sequencing platform (Illumina Hi-Seq and Illumina GA) the two separate clusters were difficult to define, implying no major contribution to the differential expression. The differential miRNA expression was analyzed using the R package, DESeq2 [45]. The expression values were normalized for sequencing depth and library composition using variance stabilizing transformation and were used for all the subsequent analysis. The significant differences in the expression between different tumor and normal controls were assumed at absolute log2 fold change ≥0.6 and adjusted *p*-value < 0.05. For clustered miRNAs, the significance of differential expression of clustered miRNAs were determined by two-proportions z-test using base R function, “prop.test”. The differential expression of the miRNA cluster was compared to a standard defined by the complement of that cluster, i.e., miRNAs not in the cluster.

### 2.4. Logistic Regression Analysis of MiR-379/656 Expression

The expression matrix was randomly divided into training data (70%; 633 primary tumors and 63 normal samples) and test data (30%; 270 primary tumors and 27 normal samples), using R package, caret [46]. Univariate binomial logistic regression models were built on training data using the base R function; “glm” and miRNAs where the *p*-value of regression model was <0.05 were removed. Then, lasso regularized logistic regression with 10-fold cross validation was performed on training data with the remaining miRNAs using R package, glmnet [47], to select lambda (λ) parameter with minimum prediction error. Final lasso regularized logistic regression model was built with lambda (λ) where the prediction error was minimum and excluded the miRNA variables with minor contributions (see results for details). The test data were used for subsequent model diagnostics. The area under curve (AUC) of receiver characteristics operating (ROC) curve was used to estimate the accuracy of the model over all possible thresholds. Confusion matrix obtained at optimal threshold was used to compute the sensitivity and specificity of the model. This process of building classification model and its subsequent evaluation were repeated 10 times with different sets of training and test data obtained via resampling. MiRNAs prioritized in 8 out 10 repetitions were further used to build separate regression models with respect to different molecular subtypes of BRCA.

### 2.5. Cox (Proportional Hazards) Regression Analysis of MiR-379/656

Principal component analysis (PCA) was performed to reduce the dimensions of MiR-379/656 expression using the base R function, “prcomp”. The first principal component (PC1), which captured maximum variation in the data, was used to represent the MiR-379/656 meta-expression (Appendix A). The meta-expression values were then examined for differences among different clinical features of breast cancer (tumor grade, tumor size/extent and molecular subtypes) using Mann–Whitney U-Test. The overlap of patients between tumor grade, tumor size/extent and molecular subtypes was determined using UpSetR plot [48]. No overlap was observed between a particular tumor grade and tumor size/extent with molecular subtype (Appendix A).

For survival analysis, patients’ follow-up information was used where overall survival (OS) was defined as the time starting from the date of diagnosis or treatment start until the time of death [49]. The association with patient survival was evaluated using univariate Cox Proportional-Hazard regression analysis. The models were adjusted for all clinical features that showed association with MiR-379/656 meta-expression. Further, Kaplan–Meier curves along with log-rank tests were used to determine statistical differences between the survival of high and low groups, defined by the median expression value as cutoff. The survival analysis was performed using R packages, survival [50,51] and survminer [52].

### 2.6. Functional Enrichment Analysis of Validated Gene Targets of MiR-379/656

The list of validated gene targets of MiR-379/656 were obtained using miRTarBase 7.0 [53]. To improve the reliability of the analysis, only 901 primary tumor samples with paired miRNA and mRNA expression were considered. Since miRNAs negatively regulate the expression of cognate target genes, anti-regulated miRNA-mRNA pairs were identified using R package, psych [54]. MiRNA-mRNA pairs with spearman correlation coefficient ≤−0.2 and adjusted *p*-value < 0.05 were shortlisted to understand the crosstalk among the deregulated pairs of miRNA-mRNA. Interaction networks were plotted using R package, ggraph [55] and tidygraph [56]. The significantly correlated mRNAs of anti-regulated miRNA-mRNA pairs were used for functional enrichment analysis using the R interface of REACTOME pathway database, ReactomePA [57] and R package, clusterProfiler [58]. Pathways with adjusted *p*-value < 0.05 were considered as significantly enriched.

### 2.7. Data Processing and Statistical Analysis

Data processing and all statistical analyses mentioned above were performed using programming software, R Foundation For Statistical Computing, Vienna, Austria (version 4.0.3). Statistical significance was assumed if *p* < 0.05. Statistical plots and other data visualizations were generated using the R package, ggplot2, unless specified otherwise [59].

## 3. Results

### 3.1. MiR-379/656 Is the Most Significant Differentially Expressed Cluster in Breast Cancer

The differential expression analysis for 1390 miRNAs (miRnome) revealed 310 upregulated (22.3%) and 176 downregulated (12.66%) miRNAs in breast cancer tumor samples compared with normal controls (Figure 1A and Appendix A). In case of MiR-379/656, 32 miRNAs (32/42; 76%) were downregulated (Appendix A). Similar observations were made when differential expression was studied in 68 paired samples (Appendix A). MiR-379/656 downregulation (confidence intervals: 51% C.I [1]–80% C.I [2]) compared with the 11% downregulation of all miRNAs outside the cluster (1348 miRNAs, used as control) revealed that the proportions were significantly different (*p* < 10^−30^) (Figure 1B). This indicated that downregulation of the MiR-379/656 cluster in breast cancer was biologically relevant and not attributable to non-specific downregulation of miRNAs.

Next, we examined if downregulation of the entire MiR-379/656 cluster was also noticeable in specific molecular subtypes of breast cancer. Differential expression analysis of each subtype against the controls revealed Basal and Luminal B to have at least 83% (35/42) of the miRNAs to be significantly downregulated (Figure 1D; Appendix A). In Her-2 and Luminal A subtypes, the downregulation of miRNAs encoded in miR-379/656 was noted at 43% and 69% respectively. Interestingly, in Her-2 subtype, MiR-323b was upregulated explaining the molecular basis of significant upregulation of MiR-323b with paired samples (Appendix A).

For a comparative and comprehensive analysis of all clustered miRNAs in breast cancer, we compared the expression of MiR-379/656 against all the miRNA clusters (miRBase V18; [40]). We found 14 clusters apart from MiR-379-656 that showed significant over-representation of dysregulated miRNAs (Figure 1C, Appendix A). MiR-379/656 was found to be the most significant differentially expressed cluster in the human genome. Amongst others, MiR17~92 on chromosome 13 has been reported as a polycistronic oncomir cluster with coordinated function to promote proliferation, increase angiogenesis and sustain cell survival [60]. The cluster showed 66% upregulation (*p* < 0.05) in our analysis and served as a positive control for the analysis pipeline (Figure 1C, Appendix A). Within chromosome 14 and within the imprinted domain DLK1-DIO3, another cluster, anti-Rtl1, also showed 75% downregulation (*p* < 10^−05^) (Figure 1C, Appendix A). Significant downregulation at a genome-wide significance level for both clusters on chromosome 14q suggested disruption of epigenomic features regulating imprinting at DLK1-DIO3 in breast cancer. The largest cluster on the human genome, the imprinted miRNA cluster on chromosome 19, with 46 miRNA encoding genes, showed non-differential expression in breast cancer (Figure 1C, Appendix A)—serving as the negative control. Thus, our analysis indicated that the potential epigenetic dysregulation for the MiR-379/656 cluster was specific and relevant for breast cancer pathogenesis and progression. In the subsequent sections, we further analyze the clinical and biological significance of this observation.

### 3.2. MiR-379/656 Accurately Classifies Tumor and Normal Samples—Especially Basal and Luminal B Subtypes

To understand the clinical relevance of MiR-379/656 downregulation, the prognostic potential of MiR-379/656 for accurate classification of tumor and normal samples was evaluated. First, using univariate logistic regression on training data, miRNAs unlikely to contribute to the pathophysiology were removed (*p* > 0.05; Appendix A). For the remaining miRNAs, most showed an association of lower expression with the disease/tumor (negative β coefficient). This was expected since the cluster was downregulated in breast cancer. After removing the miRNA variables based on the lambda value from 10-fold cross validation (Appendix A), the model prioritized 15 miRNAs in 8 out of 10 repetitions (Appendix A). The expression level of these 15 miRNAs were able to correctly identify breast cancer patients from controls with 98.43 ± 1.31% accuracy (AUC of ROC; Figure 2A). The sensitivity and specificity to distinguish breast cancer tumors from normal controls was 99.15 ± 0.58% and 77.78 ± 5.24%, respectively.

We also evaluated the prognostic potential of the subset of 15 prioritized miRNAs at MiR-379/656 for accurate classification of different molecular subtypes (Appendix A; Figure 2B). The sensitivities of each molecular subtype were 98.70 ± 1.62%, 95.10 ± 4.50%, 91.70 ± 6.24% and 56.5 ± 2.62% for Luminal A, Luminal B, Basal and Her-2, respectively. Whereas the specificity measure of Her-2, Basal, Luminal B and Luminal A was observed at 87.90 ± 1.74%, 77.00 ± 1.73%,54.80 ± 2.36% and 17.80 ± 2.36%, respectively. MiR-495 had the best trade-off between sensitivity (95.48 + 1.81%) and specificity (86.11 ± 10.64%) among all molecular subtypes of BRCA. Overall, the subset of 15 miRNAs at MiR-379/656 predicted tumor outcome with high accuracy—especially for Basal and Luminal B molecular subtypes.

### 3.3. MiR-379/656 Is Associated with Poor Clinical Outcome in Breast Cancer

We studied the association of 15 prioritized miRNA expression with different clinical features of breast cancer (Figure 3A). The meta-expression of these 15 miRNAs showed decreased expression in Basal and Luminal B subtypes of breast cancer. Further, the expression decreased with increasing tumor stage and tumor size/extent. For survival analysis, the hazard ratio was adjusted for molecular subtype, tumor size and tumor stage to negate the effect of different clinical features on patient survival. The adjusted hazard ratio (HR_adj_) of most of the 15 miRNAs in breast cancer was <1, implying that decreased expression was associated with worse patient outcome. However, the inter-tumor variability in the miRNA expression profiles revealed a wider spread (95% CI) of the hazard ratio—resulting in marginal significance for most of the candidate miRNAs (Figure 3B). Notable exceptions were miR-487a, miR-889 and miR-379. Kaplan–Meier survival plots revealed significantly worse patient outcomes associated with lower expressions for these miRNAs (miR-487a, miR-379 and miR-494; Figure 3C). These findings implicated that decreased MiR-379/656 expression was associated with poorer clinical outcome.

### 3.4. MiR-379/656 Target Genes Are Enriched for Cancer-Relevant Pathways

To understand the biological significance of altered expression of the 15 prioritized miRNAs in breast cancer, the experimentally validated mRNA targets were annotated via pathway enrichment analysis. The correlation analysis between these 15 miRNAs and their target mRNAs revealed possible miRNAs-mRNAs regulatory networks (Appendix A). Of the 15 miRNAs, 12 miRNAs were found to be negatively correlated with 103 of its validated targets across 114 interactions, among which MiR-410, MiR-889, MiR-377 and MiR-381 were identified as hub-miRNAs to interact with at least 15 mRNA targets (Figure 4A). Next, the target mRNAs were screened for enrichment of REACTOME signaling pathways. We observed 21 REACTOME signaling pathways to be enriched for the target mRNAs of MiR-379/656 (Appendix A). Cancer-related pathways, especially DNA repair, transcriptional regulation by p53 and cell cycle checkpoints were among the most significantly enriched pathways (Figure 4B). This implied that a cluster-wide downregulation of MiR-379/656 can trigger perturbations of entire mRNA/protein network and regulate oncogenic signaling pathways in a coordinated manner.

## 4. Discussion

MiR-379/656, the second largest miRNA cluster in the human genome, has been implicated in tumor progression or poor clinical outcome suggesting its potential as a cancer biomarker. In a smaller study, we had shown 46% downregulation of the cluster in 80 breast cancer tumor samples compared with controls [34]. This observation was supported by others including linking the expression of MiR-379/656 to epithelial mesenchymal transition, and stemness [39]. Candidate miRNAs from the cluster, namely, miR-127-5p, miR-544a and miR-655-3p were shown to inhibit metastasis development in an animal model of breast cancer lung colonization [38]. The data we presented in this report are the most comprehensive analysis to date, of the miR-379/656 cluster as a diagnostic and prognostic marker for breast cancer. Our unbiased genome-wide expression analysis of mRNA and miRNA in almost 1000 tumor samples shows that the cluster is profoundly dysregulated, with 76% of the miRNAs significantly downregulated in tumor samples compared with controls. In the entire dataset, there were 68 paired samples (tumor and normal tissue from the same patient), and a separate analysis of that small subset also revealed the same cluster-wide downregulation. Further, we identified a subset of 15 miRNAs that can be used for diagnostic and prognostic purposes of breast cancer and its subtypes—especially for Basal and Luminal B (83% and 86% of the clustered miRNA downregulated, respectively). As expected, molecular subtypes with greater downregulation of MiR-379/656 had a better ‘trade-off’ between specificity and sensitivity. The expression level of the 15 miRNA panel could diagnose breast cancer samples from controls with 99.15% sensitivity and 77.78% specificity (Figure 2A). Among the four main subtypes, all except Her-2 showed more than 91% sensitivity, while Her-2 showed the highest specificity (87.90%). It is possible that the 15 prioritized miRNAs were not sufficient to capture the molecular characteristics for identification of the HER2 subtype. In addition, the use of multinomial classification instead of binomial logistic regression to classify the molecular subtypes might have yielded different predictions. We chose to use the binomial logistic regression because of the unequal proportion of samples (Basal-136, Her-2-59, Luminal A-491 and Luminal B-179). This would have resulted in a ‘class imbalance’ in a multinomial model, which would eventually affect the accuracy of the prediction.

We previously reported the expression level of the MiR-379/656 cluster to be a predictor of the tumor grade for brain cancer (Gliomagenesis; [33])—where lower expression of the cluster was correlated with higher tumor grade. In the same study, we also showed that higher miRNA expression was associated with better survival. In this report, we observe the same trend with breast cancer and its subtypes. The decreased expression of the 15 miRNA panel correlated with increasing tumor stage, increasing tumor size/extent and Basal and Luminal B molecular subtypes (Figure 3A). We observed that the decreased miRNA expression was associated with worse patient outcome, albeit with marginal significance. However, a large amount of patient information was lost during follow up (right censoring), which may have impacted the survival analysis. Taken together, this necessitates further investigation using independent datasets. Our study identified MiR-495, MiR-379 and MiR-487a as potential biomarkers for breast cancer. In our analysis, MiR-495 had more than 94% sensitivity and more than 77% specificity in each molecular subtype of breast cancer. MiR-495 is reported to be dysregulated in several cancers, including breast cancer. It modulates the transition of G1 to S phase by regulating the expression of protein Bmi-1 in breast cancer [61]. The long non-coding RNA, SNHG20, regulates HER2 via miR-495 and participates in proliferation, invasion and migration of breast cancer cells [62]. MiR-495 targets STAT-3 to inhibit cell proliferation and migration and to induce apoptosis in breast cancer [63]. The observed downregulation of miR-495 can thus enhance proliferation and migration and reduce apoptosis for breast cancer cells. MiR-379 is reported to have decreased expression in breast cancer patients compared with normal controls [64]. MiR-379 expression showed positive correlation with increasing tumor stage and expression of Cyclin B1. MiR-487a is not directly implicated in breast cancer; however, the miRNA shows deregulation in other cancers [65,66]. These reports support our findings and provide interesting opportunities for the development of miRNA-based biomarkers for early detection of breast cancer.

MicroRNAs are higher order epigenetic regulators. Owing to the imperfect complementarity of the miRNA and the target mRNA sequences, individual miRNAs can bind and regulate multiple mRNAs [67]. Thus, altered expression of one or few miRNAs can alter the expression of hundreds of mRNAs, and the cascading effect can create havoc in biological systems. As shown in Figure 4B, the subset of 15 miRNAs are expected to perturb pathways relevant for cancer pathophysiology. Cluster-wide downregulation of MiR-379/656 and anti-Rtl1 cluster suggested that epigenomic features responsible for maintenance of imprinting at DLK1-DIO3 might be disrupted in breast cancer. The loss of imprinting in tumors is unexpectedly high, rendering it as one of the most common molecular mechanisms in cancer [68,69]. Corroborative evidence suggests that imprinting defects alter the expression of MiR-379/665 [70,71,72]. We have previously shown that downregulation of MiR-379/656 in glioblastoma multiforme (GBM) was consistent with hyper methylation of the locus [34]. In lung cancer, hypo methylation of DLK1-DIO3 locus has been correlated with overexpression of miRNAs encoded in the cluster [73,74]. For MiR-379/656 cluster, spanning ~45 Kb in the genome, it will be interesting to explore the possible mechanisms responsible for the said dysregulation in cancers.

This study relies on data gathered from a single source (TCGA), which may influence the findings. Thus, additional data mining and analysis are required to confirm consistency with the TCGA results, and the results are needed to be validated in cell lines and/or patient samples. In future, non-invasive or minimally invasive methods relying on liquid biopsies or circulating tumor cells can be considered for the development of MiR-379/656-based breast cancer diagnostics and/or prognostics. Future studies can determine whether MiR-379/656 and/or the cognate mRNA targets implicated in “cancer-related pathways” can be targeted for reversal of phenotypes.

## 5. Conclusions

Aberrant expression of miRNAs leads to disease pathogenesis, particularly cancer. This effect may be more pronounced for miRNA clusters. Here, we report a subset of miRNAs at the MiR-379/656 cluster, which can be used for diagnostic and prognostic purposes of breast cancer and its subtypes—especially Basal and Luminal B.

## Figures and Tables

**Figure 1 jcm-10-04071-f001:**
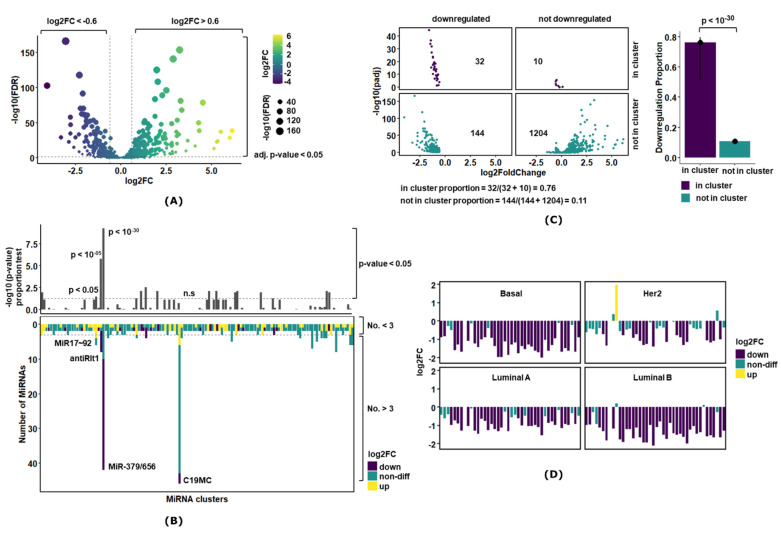
Differential miRNA expression analysis in breast cancer. (**A**) The volcano plot showing miRNA with large fold changes (*x*-axis) along with statistical significance (*y*-axis) in breast cancer. The dotted horizontal line denotes *p*-value of 0.05, and the dotted vertical lines denote absolute log2FC > 0.6. The color represents the biological significance, and size represents statistical significance. The upregulated genes are toward right and downregulated genes are toward the left of the volcano plot. (**B**) The plot represents the 2 × 2 contingency table that compares the proportion of downregulation of miRNAs in the cluster with the standard defined by its complement, i.e., downregulation of miRNAs not in the cluster. The bar-plot represents the downregulation proportions, and the vertical line on the bar represents the 95% C.I. The value above the bars represents the *p*-value of the proportion test. (**C**) The plot represents the proportion test for all 132 clustered miRNAs in the genome. The upper panel shows the *p*-value of the proportion test, and the dotted horizontal line denotes *p*-value of 0.05. The lower panel shows the distribution of differential expression of miRNAs in each cluster where the dotted horizontal represents clusters with 3 or more miRNAs. (**D**) The bar-plot showing miRNA with large fold changes along with statistical significance in molecular subtypes of breast cancer. The downregulated genes are represented by purple color, and upregulated genes toward right are denoted by yellow color.

**Figure 2 jcm-10-04071-f002:**
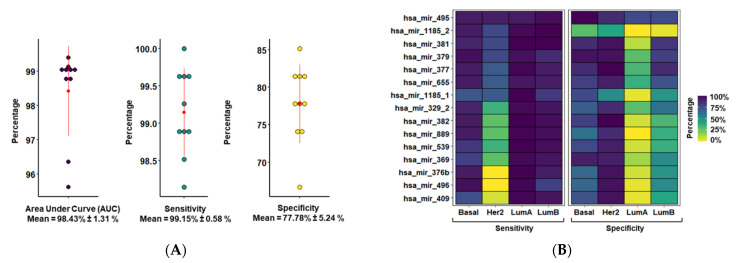
Prognostic potential of MiR-379/656 in breast cancer. (**A**) The plot shows distribution of AUC, sensitivity and specificity of classification models built using different sets of training and test data (*n* = 10). The error bar shows the mean and standard deviation. (**B**) The tile plot showing the sensitivity and specificity measure of 15 miRNAs at MiR-379/656 as determined by univariate binomial logistic regression model across molecular subtypes of breast cancer.

**Figure 3 jcm-10-04071-f003:**
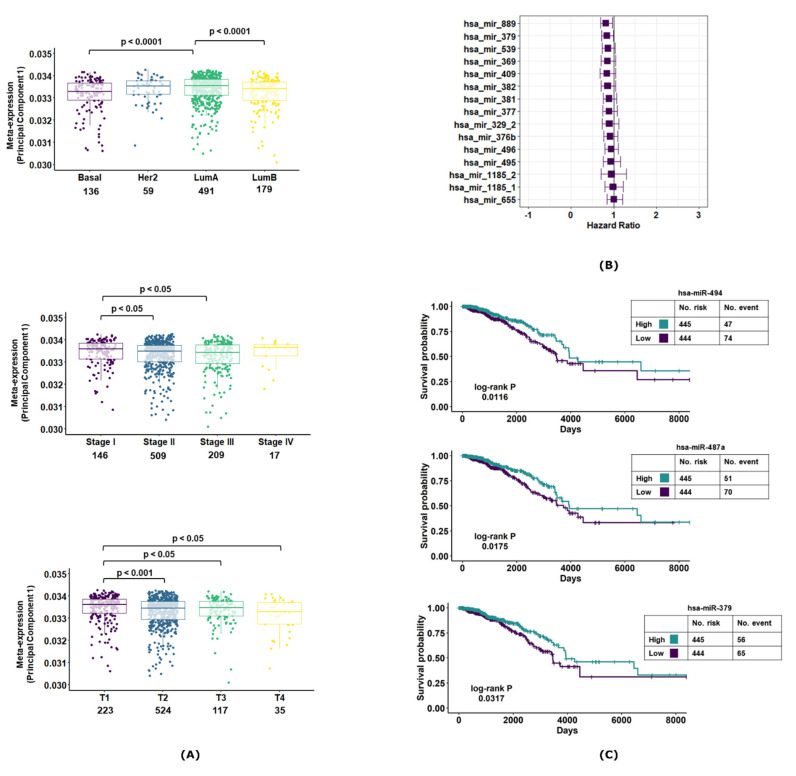
Correlation of MiR-379/656 with clinical features in breast cancer. The boxplot showing the meta-expression of 15 prioritized miRNAs at MiR-379/656 defined by the first principal component across different (**A**) molecular subtypes, tumor stages and tumor size/extent of breast cancer. The value above the boxplot represents the *p*-value of Mann–Whitney U-Test of different group comparisons. (**B**) The forest plot represents the hazard ratio along with 95% C.I. of the subset of 15 miRNAs at MiR-379/656. (**C**) The Kaplan–Meier survival curves of MiR-487a, MiR-379 and MiR-494. The patients in high- and low-expression groups are determined by the median expression value cut-off. The log-rank *p* denotes the *p*-value of the test comparing the differences in the distribution of survival in the two groups.

**Figure 4 jcm-10-04071-f004:**
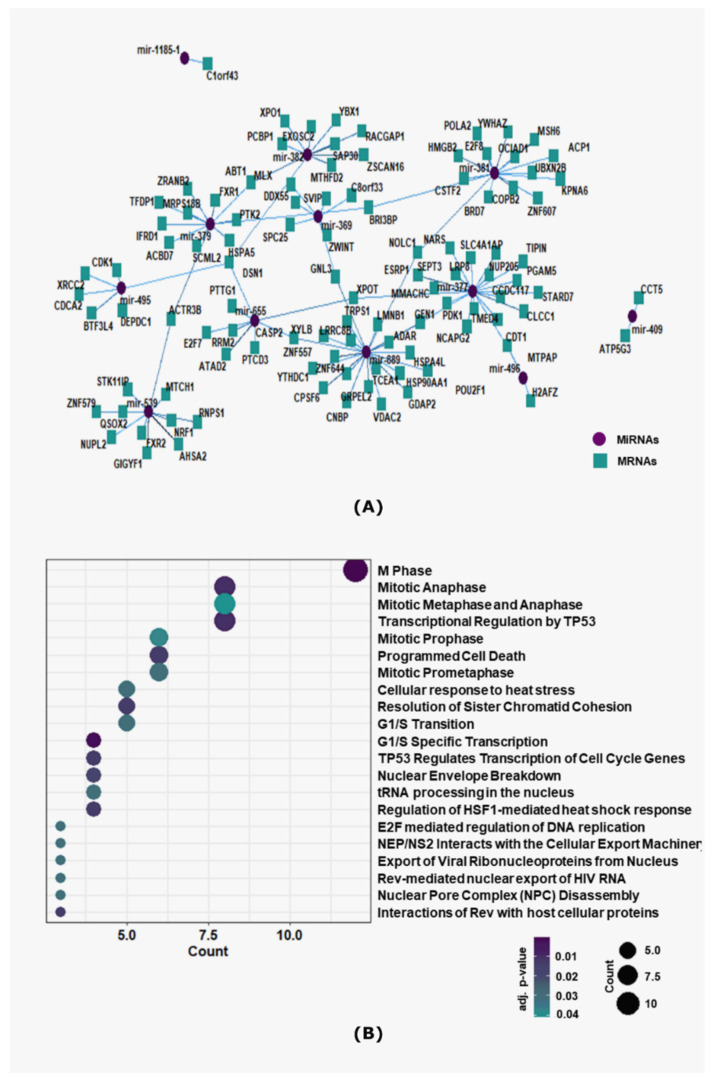
Pathway enrichment of target mRNAs of MiR-379/656 in breast cancer. (**A**) Interaction network plot of MiR-379/656 and negatively correlated target mRNAs. The purple-circled node represents the miRNAs, and the blue-squared nodes represent the mRNAs connected via edges representing interactions. (**B**) The dot-plot showing the top 20 statistically significant REACTOME pathways enriched for anti-regulated target mRNAs. Each point represents a pathway, arranged according to the overlapping count of genes, and the color of the point indicates −log10 adjusted *p*-value.

## Data Availability

All the publicly available data used in this manuscript are available to download at https://portal.gdc.cancer.gov/legacy-archive/search/f (accessed on 7 May 2021) and https://xena.ucsc.edu/ (accessed on 7 May 2021).

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
