# Peer review of "Diagnostic and Prognostic Potential of MiR-379/656 MicroRNA Cluster in Molecular Subtypes of Breast Cancer"

_jcm, 2021, doi:10.3390/jcm10184071_

Round 1
Reviewer 1 Report
Dear editor:
Thank you for inviting me to evaluate this article titled “Diagnostic and Prognostic potential of MiR-379/656 microRNA cluster in molecular subtypes of breast cancer”. This manuscript is of great interest to cancer researchers. However, there are some concerns that need to be addressed before they can be published.
Major comments:
- In this study, the authors analyzed the differential expressions of 1,390 miRNAs among 993 samples [line 103]. However, there are no descriptions of these samples in the manuscript (smoking history, age, gender, ethnicity and race, pathologic stage, sequencing platform, gene expression data filtering, etc). Some of these characteristics are fixed effects (age, gender, etc), and may confound the differentially expressed miRNAs. The authors need test whether there are any significant differences between the groups isolated by the fixed effects. In particular, some characteristics (smoking history, pathologic stage, sequencing platform, etc) will extensively affect the expression of the genes. To address the issue, the authors need to select the paired samples (the normal and tumor samples from the same patient) and identify the differentially expressed miRNAs from these paired samples. The validation and downstream analysis of these differentially expressed miRNAs can be utilized in more samples.
- The authors applied “The lasso penalized logistic regression model prioritized 24 out of 42 miRNAs by excluding the miRNA variables with minor contributions” [line 221] to classify tumor and normal samples, and obtained an accuracy of 99.18%. Therefore, they concluded that “the subset of 24 miRNAs at MiR-379/656 predicted tumor outcome with high accuracy” [line 25]. Their conclusion is unreasonable. Since the prioritized 24 miRNAs are based on the 993 samples [line 103], it is obvious that using the 23 miRNAs to classify these 993 samples [line 118] can achieve a high accuracy. But this doesn’t mean that these miRNAs will be able to correctly classify other samples. I think a more reasonable way to address the issue is by using the 993 samples as the training dataset to train the classifier, and then choosing more samples apart from the 993 samples as the test dataset to test the performance of the classifier.
- In Table S6, the “std. error” of mir-544a is extremely high compared with its “estimate”. I suspect that mir-544a might have a multimodal distribution among these samples, with each peak being a subcluster. If I am right, this may partially interpret the low sensitivity and high specificity of mir-544a in Figure 2B. The authors may analyze the association between mir-544a regulation and the performance of the classification, and the results may be very interesting.
Minor comments:
- [line 78] “MiR-379/656 on chromosome 14q32 is the second largest miRNA cluster in human (~55Kb) and harbors 42 miRNA encoding genes.” According to Nayak et al. [1], the length of the cluster might be ~45Kb. The authors need to verify the correct value and cite references to support the value.
[1] Nayak, S., Aich, M., Kumar, A. et al. Novel internal regulators and candidate miRNAs within miR-379/miR-656 miRNA cluster can alter cellular phenotype of human glioblastoma. Sci Rep 8, 7673 (2018). https://doi.org/10.1038/s41598-018-26000-8
Reviewer 2 Report
The overall data was well presented and explained well. However, few minor corrections are needed.
- In figure 1a, the authors can mark few of the most significant upregulated and diwnregulated
- In line 334 to 337, the sentence can be changed to changed for the proper understanding of the readers.
Reviewer 3 Report
The issue is interesting but I have some main remarks
First of all, I do not understand the potential diagnostic role of a miRNA (see abstract and Discussion lines 330-336)
Introduction section is too long
Results sections
Some comments have been incorporated in this section, but they must be included in the Discussion section (for example at page 4, lines 178-179); moreover, some comments are not clearly supported by the presented data (for example, at page 6, lines 255-257).
Discussion
This section is too long; authors must comment their principal findings without any digression.
Conclusions
This section must briefly underline the principal findings and future perspectives
English language must be extensively reviewed
Round 2
Reviewer 1 Report
Dear editor:
In this revision, the authors have addressed some of the previous comments. However, there are still some issues that need to be addressed before they can be published:
- [Comment 1 in my last report] The authors’ responses to this commentary are informative, but I don’t know why none of these responses were delineated in the manuscript. As part of the RNA-seq analysis, these interpretations are of particular importance. Environment (smoking history, age, gender, ethnicity and race, pathologic stage, sequencing platform, gene expression data filtering, etc) is an important factor that usually affects the expression of RNA-seq, which is why I strongly suggest using paired samples to identify the differentially expressed miRNAs. The results of paired sample analysis being similar to the overall analysis does not represent the overall analysis being logical. If the authors insist, they should at least demonstrate that the environmental factors are not significantly different between the two groups.
- [Comment 2 in my last report] [Section 3.2] “The lasso penalized logistic regression model prioritized 22 out of 42 the remaining miRNAs by excluding the miRNA variables with minor contributions”. In the entire manuscript, I cannot find any interpretations about how they prioritized 22 of the 42 miRNAs, though they answered me in the response file. Providing detailed information is important, and I think many readers will have the same confusion after reading this section.
- [Section 3.2] The accuracy of only a single classification is not convincing. The authors need to repeat their classification with different training and test datasets (resampling), and provide the mean and standard deviation for the accuracy.

Reviewer 3 Report
The Authors have partially modified their paper, that remains confused, disorganized and difficult to read
